# Synergistic Efficacy of Policosanol (Raydel^®^) and Banaba Leaf Extract to Treat Hyperglycemia and Dyslipidemia in Streptozotocin-Induced Diabetic and Hyperlipidemic Zebrafish (*Danio rerio*): Protection of Liver and Kidney with Enhanced Tissue Regeneration

**DOI:** 10.3390/ph18030362

**Published:** 2025-03-03

**Authors:** Kyung-Hyun Cho, Sang Hyuk Lee, Yunki Lee, Ashutosh Bahuguna, Ji-Eun Kim

**Affiliations:** Raydel Research Institute, Medical Innovation Complex, Daegu 41061, Republic of Korea

**Keywords:** banaba, diabetes, policosanol, hyperglycemia, fatty liver, oxidative stress, inflammation, senescence, tissue regeneration

## Abstract

**Background:** The efficacy of banaba leaf extract was tested against carboxymethyllysine (CML)-induced toxicity in embryos and adult zebrafish. Additionally, the individual and combined effects of banaba (BNB) and policosanol (PCO) were analyzed to alleviate dyslipidemia, hyperglycemia, and associated effects in streptozotocin (STZ)-induced hyperlipidemic diabetic zebrafish. **Methodology:** The high cholesterol diet (HCD, final 4%, *w*/*w*)-fed zebrafish were injected with STZ to develop diabetes and were subsequently fed with either HCD or HCD+BNB (final 0.1% *w*/*w*) or HCD+PCO (final 0.1% *w*/*w*) or HCD+BNB+PCO (each final 0.1%, *w*/*w*) each for 14 days. The zebrafish tail fin was amputated to assess tissue regeneration, while the organs and blood were collected for histological and biochemical analysis. **Results:** Severely compromised embryo survivability and developmental defects were noticed in the CML-injected group that significantly improved following BNB exposure. Similarly, CML-induced acute paralysis and mortality of adult zebrafish were effectively mitigated by the treatment with BNB. In the hyperlipidemic diabetic zebrafish, both BNB and PCO supplementation displayed the hypoglycemic effect; however, a remarkable reduction (*p* < 0.05) in blood glucose levels was observed in the BNB+PCO group, around 14% and 16% less than the BNB group and PCO group, respectively. Likewise, higher tail fin regeneration was noticed in response to BNB+PCO supplementation. Both BNB and PCO have a substantial counter-effect against HCD+STZ-induced dyslipidemia. However, the combined supplementation (BNB+PCO) displayed a significantly better effect than that of BNB and PCO alone to alleviate total cholesterol (TC), triglycerides (TGs), and low-density lipoprotein cholesterol (LDL-C). The most impressive impact of BNB+PCO was noticed in the elevation of high-density lipoprotein cholesterol (HDL-C), which was ~1.5 times higher than the HDL-C level in response to BNB and PCO. Also, BNB+PCO effectively reduced the malondialdehyde (MDA) and elevated the plasma sulfhydryl content, paraoxonase (PON), and ferric ion reduction (FRA) activity. Histological analyses revealed a significant effect of BNB+PCO in preventing inflammatory infiltration, fatty liver changes, and interleukin-6 production. Similarly, a notably better effect of BNB+PCO compared to their individual effect was noticed in preventing kidney damage and mitigation of ROS generation, apoptosis, and cellular senescence. **Conclusions:** The finding establishes the substantial effect of BNB and PCO in countering hyperglycemia, dyslipidemia, and associated disorders, which synergistically improved following the combined supplementation with BNB+PCO.

## 1. Introduction

Diabetes is a metabolic disease characterized by hyperglycemia which is spreading across the globe, with the alarming rate reflected by 463 million diabetic patients in 2019, predicted to reach over 700 million by 2045 [1,2]. The sedentary lifestyle, poor nutrition, obesity, smoking, and aging are among the important factors contributing to the onset of diabetes [3,4]. Type 2 diabetes is the most predominant form of diabetes, arising from the combination of insulin resistance and impaired insulin secretion due to the dysfunction of pancreatic β-cells [5]. Diabetes has a detrimental effect on multiple organs, leading to a morbid state like retinopathy, vasculopathy, neuropathy, nephropathy, and cardiovascular diseases [6]. Patients with diabetes are prone to coronary artery disease, myocardial infarction, and cardiovascular diseases compared to non-diabetic individuals [7]. It has been reported that about 12% of diabetic individuals present with diabetic cardiomyopathy attributed to myocardial fibrosis and myocellular hypertrophy and are at a higher risk of heart failure [7,8]. In addition, a substantial adverse effect of diabetes has been noticed with respect to the induction of fatty liver disease, liver fibrosis, and cirrhosis [6].

The hyperglycemic environment exacerbates hepatocyte damage in fatty liver disease and, consequently, augments the morbidity and mortality of diabetic patients [6,9]. Furthermore, diabetes adversely affects the normal tissue regeneration process, leading to varied complications such as diabetic foot ulcers [10]. The elevated glycemic condition also leads to the production of a variety of advanced glycation end-products (AGE) [11] such as carboxymethyllysine (CML) (chronic wound+CML). The elevated AGE upregulates the receptor for advanced glycation end-product (RAGE) expression in the pancreatic β-cells and the AGE-RAGE interaction induces nuclear factor-kB (NF-kB) signaling, leading to the intense inflammation and programmed cell death that ultimately affect the pancreatic β-cells [11].

Insulin resistance and hyperglycemia substantially elevate blood pressure and provoke dyslipidemia [2], highlighted by elevated blood total cholesterol (TC), triglycerides (TGs), low-density lipoproteins (LDL-C), and decreased high-density lipoproteins (HDL-C) levels, thereby amplifying the risk of cardiovascular complications associated with diabetes [7]. There are many synthetic drugs of different classes used to prevent diabetes and dyslipidemia. However, the toxicity associated with their long-term consumption is always a matter of concern [12] and highlights the significance of natural compounds in therapeutic applications [12]. Natural compounds and alternative therapies are gaining attention owing to their affordability and safety over conventional pharmaceutical treatments [12]. Furthermore, when used in combination with established therapies, natural compounds have demonstrated substantial potential in effectively managing the advanced stage of type 2 diabetes [12,13]. Several natural compounds and extracts have been explored as an alternative therapy to mitigate diabetes [12] and dyslipidemia [14].

Among the natural compounds, policosanol, which is a typical mixture of long-chain aliphatic alcohols (LCAA), has gained huge attention due to its diverse functionality including antiglycation and tissue regeneration activity [15,16]. However, the prominent effect of policosanol has been observed to counter dyslipidemia [17] and hypertension [18]. Several preclinical and clinical studies have documented the impactful role of policosanol in reducing TC, TGs, and LDL-C levels as well as augmenting HDL-C levels [17,19] via the inhibitory impact on the cholesterylester transfer protein (CETP) [16,20]. Also, policosanol prevents LDL oxidation, augments the cholesterol efflux ability, and boosts the HDL-C functionality through the activity enhancement of HDL-associated paraoxonase (PON)-1 [21]. In a comparative study, policosanol displayed a better effect than lovastatin (a conventional statin drug) on the elevation of HDL-C levels and reduced the LDL/HDL ratio in human subjects of type 2 diabetes and hypercholesterolemia [22]. The effect of policosanol in preventing CML (an important AGE)-induced toxicity has been described [18], indicating that policosanol may ameliorate the hyperglycemic-induced events.

Banaba (*Lagerstroemia speciosa*), a perennial medicinal plant from the Lythraceae family, is widely cultivated across South Asian countries and has long been utilized in traditional medicine for its therapeutic properties [23]. The reports highlight the analgesic, antimicrobial, anti-obesity, antioxidant, anti-inflammatory, antifibrotic, and antihypertensive properties of banaba leaves [24]. However, banaba leaves and their important phytoconstituents, e.g., corosolic acid [23], have been extensively studied for antidiabetic properties by improving glucose metabolism, insulin secretion, insulin sensitization/resistance, and inhibition of enzymes like α-amylase and α-glucosidase [23,25]. Several studies have observed the impactful role of banaba and corosolic acid in decreasing the blood glucose levels in genetically diabetic and chemically induced diabetic animals [23]. Corosolic acid exerts its antidiabetic effects by regulating insulin receptor substrate-1 (IRS-1), protein kinase B (Akt), and liver kinase B1 (LkB1), thereby mitigating adenosine monophosphate-activated protein kinase (AMPK) activation to alleviate insulin resistance [26]. Also, IRS-1 modulates phosphoinositide 3-kinase (PI3K), facilitating Akt/mTOR pathway activation, which influences the translocation of glucose transporter type 4 (GLUT4) and leads to the absorption of glucose [26]. Furthermore, corosolic acid reduces cyclic adenosine 3′,5′-monophosphate (cAMP) levels and inhibits protein kinase A activity, leading to decreased fructose-2,6 diphosphate and suppression of gluconeogenesis [23,26]. Despite several beneficial effects, the banaba has not been probed for diabetic tissue regeneration and AGE-triggered toxicity.

Given banaba substantial antidiabetic properties and the established efficacy of policosanol in managing dyslipidemia, we hypothesize that their combination might effectively mitigate the chronic complications of diabetes and hyperlipidemia. Concerning this, the present study investigates the effect of banaba on CML-induced toxicity in embryos and adult zebrafish (*Danio rerio*). Additionally, the comparative tissue regeneration effect of banaba and policosanol alone or in combination (banaba+policosanol) is assessed on streptozotocin (STZ) and high cholesterol diet (HCD)-induced diabetic and hyperlipidemic zebrafish. Furthermore, the effect of these treatments on the blood oxidative and antioxidant variables, lipid profiles, and liver and kidney functionality of diabetic hyperlipidemic zebrafish are evaluated.

## 2. Results

### 2.1. Zebrafish Embryo Protective Effect of Banaba

Figure 1A illustrates that survivability was significantly influenced by time (F = 2130.1, *p* < 0.001), treatment type (F = 3237.9, *p* < 0.001), and their interaction (F = 292.6, *p* < 0.001), indicating the influence of both factors on survivability. The survivability of zebrafish embryos was severely compromised in the CML-injected group. A periodic sharp decline in the survivability of embryos was observed in the CML-injected group, starting from 57% 5 h post injection, reaching 12% 24 h post injection, and, finally, ending with a mere 4% survivability 72 h post injection (Figure 1A). In contrast, banaba co-injection at both concentrations (5 ng and 10 ng) effectively protected the embryos from the CML-triggered mortality. The 5 ng and 10 ng banaba-treated groups showed a 61% and a 65% embryo survivability 72 h post injection, a level which was 15.2 (*p* < 0.001) and 16.2 (*p* < 0.001) times higher than the survivability of embryos observed in the CML-injected group.

The morphological analysis revealed severe developmental defects (indicated by the blue arrow) and delayed embryo hatching (indicated by the green arrow) in response to CML injection (Figure 1B). At 48 h post injection, only 10% hatching was observed among the embryos who survived in the CML-injected group, such a percentage increasing to 23.3% and 30% when CML was co-injected with 5 ng and 10 ng of banaba extract, respectively. Notably, most of the surviving embryos appeared to have severe developmental deformities (indicated by the blue arrow) concerning the tail fin curvature, the yolk sac, and the pericardial edema in the CML-injected group. In contrast, CML-induced developmental defects were prevented by the injection of banaba. However, the occasional tail fin curvature and yolk sac edema were noticed in the banaba-treated groups.

DHE and AO staining revealed massive ROS production and apoptosis in the embryos exposed to CML. The CML-induced ROS and apoptosis in the embryos were significantly protected by the co-injection of 5 ng and 10 ng of banaba extract. As depicted in Figure 1C–E, a ~five-fold (*p* < 0.001) and ~2.5-fold (*p* < 0.001) decrease in DHE and AO fluorescence intensity was quantified in the banaba extract (5 ng and 10 ng) group compared to the CML-injected groups, attesting a substantial effect of the banaba extract to ameliorate the CML-induced toxicity promoted by ROS-mediated apoptotic embryo death.

### 2.2. Banaba Mitigates CML-Induced Paralysis and Mortality in Adult Zebrafish

The statistical outcomes derived from the two-way ANOVA, as depicted in Figure 2A,B, demonstrate that treatment type (F = 202.9, *p* < 0.001 and F = 533.1, *p* < 0.001), time (F = 68.8, *p* < 0.001 and F = 135.4, *p* < 0.001), and their interactions (F = 19.9, *p* < 0.001 and F = 38.2, *p* < 0.001) significantly influenced zebrafish survivability and swimming recovery. These findings highlight the impact of both treatment conditions and time in determining zebrafish survivability and swimming recovery. In the PBS-injected group, a 96.7% survivability was observed 90 min post injection. In contrast, the severe impact of the CML injection on zebrafish survivability was noticed (Figure 2A), where survivability declined to 43.3% 30 min post injection and eventually reached 23.4% 90 min post injection. Contrary to this, the co-injection of banaba effectively prevented CML-induced mortality, reflected by a 63.3% and 56.7% survivability 60 and 90 min post injection that was ~2.4 times (*p* < 0.001) higher than the survivability observed in the CML-injected group at the respective time.

A severe paralytic effect of the CML injection was noticed in the zebrafish. As reflected in Figure 2B,C, all the zebrafish were lying at the bottom of the tanks without any sign of movement 30 min post injection in the CML-injected group, contrary to 96.7% of the zebrafish swimming in the PBS-injected group. With the progression of time (at 90 min), a slight recovery was observed, with 10.0% of the zebrafish swimming in the CML-injected groups. In contrast, the banaba extract effectively reverted the CML-provoked paralysis, as reflected by the 20% and 50% restoration of zebrafish swimming behavior, values that are approximately three times and five times (*p* < 0.001) higher than the zebrafish swimming activity noticed in the CML-injected group 30 and 90 min post injection.

### 2.3. Banaba Extract Augmented CML-Impaired Plasma Antioxidant Variables

An elevated MDA level was quantified in the plasma of the CML-injected group which was 2.2 times (*p* < 0.01) higher than the basal level detected in the PBS-injected group (Figure 3A). In response to the injection of banaba, a significant 32.8% (*p* < 0.05) reduction in the MDA level was observed compared to the CML-injected group.

The plasma sulfhydryl content was significantly reduced, by a factor of 1.6, in the CML-injected group compared to the basal level as detected in the PBS group (Figure 3B). The CML-diminished sulfhydryl content experienced a significant increase of 31.1% (*p* < 0.01) following the injection of banaba, attesting to the positive effect of the banaba extract on the plasma sulfhydryl content. Likewise, reduced plasma ferric ion reduction ability (FRA) (Figure 3C) and paraoxonase (PON) activity (Figure 3D) were noticed in the CML-injected group, such a decrease turning into a significant increase of 12.4% (*p* < 0.05) and 78.1% (*p* < 0.01), respectively, in the banaba-injected group. The combined results underline the effective role of the banaba extract in improving plasma oxidative and antioxidant variables.

### 2.4. Banaba and Policosanol Improved STZ-Impaired Tail Fin Regeneration

The amputated tail fin regenerative effect of banaba and policosanol alone or in combination was assessed in a time-dependent manner under the influence of diabetic hyperlipidemic conditions induced by HCD+STZ (Figure 4A,B). The statistical outcomes (two-way ANOVA), as depicted in Figure 4B, reveal that time (F = 122.7, *p* < 0.001), treatments (F = 268.8, *p* < 0.001), and their interactions (F = 13.7, *p* < 0.001) had a significant effect on the amputated tail fin generation, attesting the importance of both time and treatments on tail fin regeneration. The smallest degree of tail fin regeneration (9.5 ± 0.3 mm^2^) was observed in the HCD+STZ group, with values 1.8 times (*p* < 0.001) lower than the tissue regenerative area (16.6 ± 0.8 mm^2^) observed in the HCD group after 14 days, underscoring the adverse effect of STZ on the tail fin regeneration. The consumption of banaba and policosanol had a substantial curative impact on the STZ-impaired tissue regeneration, reflected by a 40.1% (13.3 ± 0.4 mm^2^, *p* < 0.001) and 34.2% (12.7 ± 0.3 mm^2^, *p* < 0.001) higher regenerated tissue area compared to that in the HCD+STZ group (9.5 ± 0.3 mm^2^) after 14 days. The combined effect of banaba and policosanol (BNB+PCO) showed the most promising outcome, evident through a significant 54.6% (14.7 ± 0.3 mm^2^, *p* < 0.001) higher tissue regeneration compared to that in the HCD+STZ group and a significant 15% and 10% increase in tail fin regeneration compared to the effect observed in the banaba only and policosanol only groups, respectively. Notably, the frequent presence of tissue fissures (Figure 4A, indicated by the blue arrows) was observed in the regenerated tail fin of the HCD+STZ group, such fissures completely disappearing in the banaba, policosanol, and banaba+policosanol groups, strengthening the significant tissue regenerative effect of banaba and policosanol alone or in combination against the STZ-posed adversity.

### 2.5. Banaba and Policosanol Displayed a Hypoglycemic Effect

An elevated blood glucose level (81.8 ± 3.6 mg/dL) was observed in the STZ-injected group which was 1.7 times (*p* < 0.001) higher than the basal glucose level (48.7 ± 2.0 mg/dL) detected in the HCD group (Figure 4C). The STZ-induced elevated blood glucose level was significantly reduced to 21.0% and 18.8% following the consumption of banaba (64.8 ± 1.6 mg/dL, *p* < 0.01) and policosanol (66.4 ± 2.5 mg/dL, *p* < 0.01). The combination of banaba+policosanol emerged with the lowest glucose level (56.1 ± 1.9 mg/dL) that was, notably, 32% (*p* < 0.001) lower than the glucose level quantified in the STZ-injected group and 14% (*p* < 0.05) and 16% (*p* < 0.05) lower than the individual effect of banana and policosanol, respectively.

### 2.6. Blood Lipid Profile

The total cholesterol (TC, 272.2 ± 18.4 mg/dL), triglyceride (TG, 49.2 ± 7.2 mg/dL), high-density lipoprotein cholesterol (HDL-C, 19.2 ± 5.0 mg/dL), and low-density lipoprotein cholesterol (LDL-C, 243.3 ± 16.9 mg/dL) levels in the STZ-injected group non-significantly (*p* > 0.05) changed compared to the only HCD-consuming group (Figure 5A–D). The consumption of banaba and policosanol significantly diminished the HCD+STZ elevated blood TC, TG, and LDL-C levels. In contrast, banaba, in combination with policosanol (BNP+PCO), displayed a much stronger effect on the reduction of HCD+STZ-altered TC, TG, and LDL-C levels. Significantly lower levels of TC, by 34.2% (179.2 ± 12.6 mg/dL, *p* < 0.01), TGs, by 59.1% (20.1 ± 3.2 mg/dL, *p* < 0.05), and LDL-C, by 67.9% (77.9 ± 15.9 mg/dL, *p* < 0.05) were quantified in the BNB+PCO-consuming group compared to the HCD+STZ group. Both policosanol and banaba increased the HDL-C level by a factor of three (*p* < 0.001) compared to the HCD+STZ groups. However, the most notable effect of a 5.1-fold (*p* < 0.001) augmentation in the HDL-C level was observed as a result of the combination of banaba and policosanol compared to the HCD+STZ groups. Compared to policosanol and banaba supplementation, ~1.5 times (*p* < 0.01) higher HDL-C levels were observed in the BNP+PCO-consuming group.

### 2.7. Effect of Banaba and Policosanol Consumption on Blood Antioxidant Status

A significant 1.6-fold increase in the MDA level was detected in the blood of individuals in the HCD+STZ group compared to the only HCD-consuming group, indicating the adverse effect of STZ on the plasma lipid peroxidation (Figure 5E). Consumption of policosanol and banaba alone or in combination significantly reduced HCD+STZ-induced MDA levels by 32.8% (*p* < 0.05), 26.7% (*p* < 0.05), and 35.7% (*p* < 0.01), respectively.

The HCD+STZ treatment substantially diminished the plasma sulfhydryl content, which was ~1.6 times (*p* < 0.001), 1.4 times (*p* < 0.01), and 1.6 times (*p* < 0.001) higher in response to the consumption of policosanol, banaba, and the combination of banaba and policosanol, respectively (Figure 5F).

A significant 10.5% reduction in FRA activity was observed in the blood of the HCD+STZ group compared to the HCD only group (Figure 5G). PON activity in the HCD+STZ group was increased by 18.6%, 24.4%, and 55.5% following the consumption of policosanol, banaba, and the combination of policosanol with banaba, respectively. Similarly to FRA activity, ~1.8 (*p* < 0.05), 1.5 (*p* < 0.05), and 1.8 times (*p* < 0.001) higher PON activity was observed in the plasma obtained from the groups consuming policosanol, banaba, and banaba+policosanol, respectively, compared to the HCD+STZ groups (Figure 5H).

The combined results reflect the fact that banaba and policosanol supplementation have a substantial protective effect against the HCD+STZ-induced oxidative effect and the restoration of plasma antioxidant status.

### 2.8. Hepatic Histology and Inflammation

The histology outcomes, as exemplified in Figure 6, display substantial hepatic degeneration and neutrophil infiltration in the HCD+STZ group. The HCD+STZ-induced hepatic changes were significantly prevented by the consumption of banaba and policosanol, as indicated by ~two-fold (*p* < 0.05) decrease in neutrophil counts compared to the HCD+STZ group. Compared to the individual supplementation with banaba or policosanol, the combination of both showed a significantly better hepatoprotective effect marked by a 3.7-fold (*p* < 0.01) and 1.6-fold (*p* < 0.05) decrease in neutrophil counts compared to that detected in the HCD+STZ group and in the policosanol- and banaba-consuming groups, respectively.

The IHC staining reflects the higher IL-6 production rate in the HCD+STZ group which accounts for 11.5% of the IL-6-stained area (Figure 6 C,D,G). In contrast to this, a reduced IL-6-stained area in the hepatic section of banaba- (6.5%) and policosanol-supplemented (6.3%) individuals was noticed that accounted for a ~1.8-fold (*p* < 0.001) reduction in the IL-6-stained area compared to that of the HCD+STZ group, indicating a substantial antiinflammatory effect of banaba and policosanol. Compared to the individual supplementation, the combined supplementation with banaba and policosanol displayed a higher reduction of IL-6 production, reflected by a 4.4% IL-6-stained area that was 2.6 times (*p* < 0.001) lower than the IL-6 production observed in the HCD+STZ group and ~1.5 times (*p* < 0.001) lower than the IL-6-stained area observed in the banaba- and policosanol-consuming groups.

The ORO staining revealed fatty liver changes indicated by a high lipid accumulation in the HCD+STZ-consuming group which was, notably, 1.4 times (*p* < 0.001) higher than the ORO-stained area of the only HCD-consuming group (Figure 6E,H). The HCD+STZ-provoked fatty liver changes were effectively curtailed by the consumption of banaba and policosanol, evident by a two-fold (*p* < 0.001) and a 1.8-fold (*p* < 0.001) decrease in lipid accumulation compared to the HCD+STZ group. However, the combined supplementation with banaba+policosanol showed a much stronger protective effect, reflected by a ~two-fold (*p* < 0.001) decrease in the ORO-stained area compared to the policosanol- and banaba-supplemented groups and a 3.6-fold (*p* < 0.05) decrease in the ORO-stained relative to the HCD+STZ group.

### 2.9. ROS Production, Apoptosis, Senescence, and Hepatic Function Biomarkers

DHE staining revealed severe ROS production in the HCD+STZ group that was effectively reduced by a factor of 1.6 (*p* < 0.001) and 1.5 (*p* < 0.001) following the supplementation with banaba and policosanol (Figure 7A,D). Furthermore, a more beneficial effect with respect to the inhibition of ROS generation was observed in response to the combined supplementation with banaba and policosanol, evidenced by a significant 23.3% (*p* < 0.01) and 31.4% (*p* < 0.01) decrease in DHE fluorescence intensity compared to the banaba- and policosanol-supplemented groups, respectively.

A similar trend was observed for apoptosis, where the HCD+STZ treatment induced a higher AO fluorescence intensity, which was effectively reduced by a factor of 1.6 (*p* < 0.001), 1.5 (*p* < 0.001), and 2.4 (*p* < 0.001) following the supplementation with policosanol, banaba, and the combination of banaba+policosanol, respectively (Figure 7B,E). Compared with the effect of banaba and policosanol alone, a 32.2% and a 37.7% decrease in AO fluorescence intensity was observed relative to the banaba+policosanol-supplemented group.

The SA-β-gal staining showed a higher abundance of senescent positive cells (%) in the HCD+STZ group that significantly (*p* < 0.001) decreased to 22.8%, 29.8%, and 38.9% following supplementation with banaba, policosanol, and the combination of banaba+policosanol, respectively (Figure 7C,F). By comparing policosanol, banaba, and their combination, the banaba+policosanol-supplemented group demonstrated a 21.7% and a 14.0% reduction in SA-β-gal-positive cells compared to the banaba and policosanol groups, respectively.

The STZ-induced increase in the blood hepatic function biomarkers AST and ALT substantially decreased following supplementation with banaba, policosanol, and their combination. A 17% and a 19% reduction in AST and a 27% and a 30% reduction in ALT levels were observed in the banaba- and policosanol-supplemented groups, respectively (Figure 8). Contrary to the individual supplementation, the combined supplementation with banaba+policosanol displayed the greatest reduction of AST and ALT levels, amounting to a 35% and a 44% reduction compared to the STZ-injected group.

### 2.10. Histological Analysis of the Kidney

As exemplified in Figure 9A, highly disorganized distal and proximal tubules with the frequent appearance of enlarged tubular lumen and luminal debris were noticed in the kidney section of the HCD+STZ group. The supplementation with banaba and policosanol effectively prevented the HCD+STZ-induced nephrotoxicity and restored the disturbed distal and proximal tubular arrangement; however, the occasional presence of increased luminal tubules with tubular debris was noticed. Conversely, the combined effect of banaba+policosanol supplementation prevented the HCD+STZ-triggered kidney damage, as shown by the regular distribution of proximal and distal tubules, which were mostly free from the cellular debris in the tubular cast.

The DHE (Figure 9B,E) and AO (Figure 9C,F) staining revealed the highest red and green color fluorescence intensity corresponding to the ROS production and apoptosis in the kidney section of the STZ+BNB group. The supplementation with both policosanol and banaba protected the kidney from excessive ROS generation and apoptosis, evidenced by a 39% and a 31.7% reduction in DHE and a 40.1% and a 33.9% reduction in AO fluorescence intensity in the banaba- and policosanol-consuming groups, respectively, compared to the STZ+BNB group. However, the combined supplementation with banaba+policosanol emerged as having a stronger impact, reflected by a 23.3% and a 31.1% reduction in DHE and a 33.3% and a 37.7% reduction in AO fluorescence intensity compared to the fluorescence intensities quantified in the policosanol- and banaba-consuming groups, respectively.

Similarly, the HCD+STZ-induced cellular senescence was prevented by the consumption of banaba and policosanol (Figure 9D,G). Contrary to the individual supplementation, the combined supplementation with banaba and policosanol displayed a ~22.9% (*p* < 0.01) reduction in senescence-positive cells compared to the banaba- and policosanol-supplemented groups and a 36.8% (*p* < 0.01) reduction in SA-β-gal-positive cells compared to the HCD+STZ group. The study outcome confirms the superior effect of the combination of banaba and policosanol over their individual effect to protect the HCD+STZ-induced nephrotoxicity. 

## 3. Discussion

CML is a typical advanced glycation end-product (AGE) and is well recognized to induce several detrimental effects [27,28] through the induction of oxidative stress and inflammation. Similarly, we observed high ROS generation levels in the CML-injected embryos that led to the teratogenic effect and apoptotic cell death of the embryos. The CML-induced ROS generation and apoptosis were substantially inhibited by the exposure to banaba, consequently preventing developmental deformities and rescuing the embryos from CML-mediated death. Results endorsed by earlier studies document the strong antioxidant properties of banaba and its phytoconstituents that directly scavenge the free radicals [29] and modulate cellular antioxidants through the activation of Nrf-2 [30], an important signaling molecule that regulates the expression of antioxidants like superoxide dismutase (SOD), catalase, reduced glutathione (GSH), and glutathione S-transferase (GST) [31]. Not restricted to embryos, the severe CML-induced toxicity to adult zebrafish was substantially undermined by the exposure to banaba, highlighting the strong curative effect of banaba which can rescue the zebrafish from the CML-arrested swimming activity and CML-induced mortality. Probably, the strong antioxidant and anti-inflammatory action of banaba is behind the protection of adult zebrafish from CML-triggered paralysis. This postulation is supported by earlier findings describing an association between acute inflammation and paralytic disorders [32]. As the banaba plays a substantial role in curtailing the generation of proinflammatory IL-6 [24] and other inflammatory markers [24,33], it consequently rescues the zebrafish from the acute CML-imposed paralysis. This finding aligns with previous studies that have deciphered the effect of tocilizumab (an IL-6 inhibitor) on restoring zebrafish swimming ability impaired by the administration of CML [32].

MDA is an important lipid peroxidation marker that documents the status of oxidative stress [34]. Similarly, the plasma sulfhydryl content is an important stress marker [35,36], and its diminished level has been associated with a variety of disease conditions [37]. CML-impaired MDA levels and plasma sulfhydryl content are effectively restored by the exposure to banaba, attesting the protective effect of banaba on CML-induced oxidative stress. Consistently, the elevated plasma FRA and PON activity in response to banaba attests the significant role of banaba in boosting the antioxidant status and is consistent with the accumulating literature describing the impact of banaba on the antioxidant variables [24]. However, to the best of our knowledge, this is the first report describing the effect of banaba on PON functionality, an important antioxidant associated with HDL, a versatile molecule with a multifaceted role including antioxidant and anti-inflammatory properties [38]. The results of this study, performed on embryos and adult zebrafish, highlight the protective effect of banaba in alleviating CML-induced oxidative stress.

Encouraged by these promising findings, the study was expanded to evaluate the effects of banaba on hyperlipidemic and hyperglycemic (diabetic) conditions induced by HCD+STZ in zebrafish. Additionally, the efficacy of banaba was compared with that of policosanol and their combination to identify the most effective formulation for mitigating chronic pathological conditions in the hyperlipidemic diabetic zebrafish. Supplementation with HCD is an efficient way to cause dyslipidemia [39]; likewise, STZ-induced type 2 diabetes, achieved through the reduction of insulin secretion from the pancreatic β-cells and its impact on insulin resistance [40], leads to hyperglycemia and associated disorders in a variety of model organisms [41,42], including the zebrafish [43]. An elevated glucose level was detected in the HCD+STZ group, such an increase being substantially prevented through the consumption of banaba, policosanol, and the combination of banaba+policosanol, attesting the impact of these supplements in protecting the organisms from STZ-induced hyperglycemia. AMP-activated protein kinase (AMPK) and phosphatidylinositol 3-kinase/protein kinase B (PI3K/Akt) are important signaling pathways that influence insulin secretion and enhance insulin sensitization, thus playing a critical role in diminishing hyperglycemic conditions. Fortunately, policosanol has a positive impact on these pathways [44] and, thus, effectively controls STZ-induced hyperglycemia. Similarly, banaba has a notable effect on glucose metabolism and insulin sensitization [23], leading to hypoglycemic effects. The effect of corosolic acid (an important component of banaba) on the activation of the AMPK signaling pathway [26], protein kinase B (Akt), and insulin receptor substance (IRS)-I phosphorylation [26,45] has been recognized as an important event modulating the insulin signal transduction, thus having a hypoglycemic effect. In addition, corosolic acid plays a role in inhibiting α-glucosidase [26,46] and α-amylase [26] enzymatic activity, consequently affecting the post-prandial glucose level. Compared to the individual supplementation, the combination of banaba+policosanol exhibited a markedly enhanced hypoglycemic effect, potentially due to their synergistic interaction that modulates the protective molecular events effectively, leading to an improved glycemic control in the zebrafish subjected to HCD+STZ-induced hyperglycemia. Despite the numerous essential developmental processes, such as glucose metabolism, which are comparable to those observed in humans [47], zebrafish differ in many aspects, for example in their insulin regulation and pancreatic function not being identical to that of humans. Nonetheless, it is evident from the present preclinical findings that the BNB+PCO combination has a substantial protective effect against STZ-triggered diabetic conditions in hyperlipidemic zebrafish and may elicit a similar response to that of human subjects. However, a clinical study needs to be conducted to establish the efficacy of BNB+PCO in dealing with diabetic conditions in human subjects.

The effects of banaba and policosanol, both of which were substantially and synergistically improved by their combination, were noticed in relation to the amputated tail fin regeneration and glucose lowering effect under the chronic hyperlipidemic and diabetic conditions imposed by HCD+STZ in the zebrafish. There has been no study, so far, describing the effect of banaba on wound healing and tissue regeneration; however, some preliminary studies have documented the tissue regenerative effect of policosanol against CML-induced chronic wounds [15,16]. The substantially lower glucose level associated with the combination of banaba+policosanol compared to that associated with their individual supplementation is the key contributor to the better tail fin regeneration effect. The notion is in accordance with reports documenting high-glucose environments leading to intense inflammation and having an impact on the vascular endothelial growth factor tissue regeneration and delayed wound healing [48,49]. In addition, hyperglycemia led to the protein glycation resulting in the formation of the AGE product, which leads to ROS generation and elevated inflammation via the nuclear factor kappa-light-chain-enhancer of activated B cells (NFκB) and increases apoptosis in fibroblasts and keratinocytes, leading to the inhibition of tissue repair processes [50]. Beyond its hypoglycemic properties, banaba exhibits significant antioxidant and anti-inflammatory properties [24], while policosanol demonstrates a potent antiglycation effect and tissue regeneration effect [15,16]; together, these events work synergistically to ameliorate HCD+STZ-induced adversities, thereby prompting an enhanced tissue regeneration. The positive effect of corosolic acid (an important component of banaba) [51] and policosanol [17,18] has been noticed against dyslipidemia; similarly, we noticed a significant reduction in TC, TGs, and LDL-C in response to banaba and policosanol. Earlier, it has been described that hexacosanol (a major LCAA of policosanol) modulates the activity of 3-hydroxy-3-methyl-glutaryl-coenzyme A reductase (HMG-CoA), a main rate-limiting enzyme of cholesterol biosynthesis [52]. In contrast to banaba and policosanol alone, the combination of BNB+PCO elicited a significantly larger reduction of the HCD+STZ-elevated TC, TG, and LDL-C levels, suggesting that the synergisms between policosanol and banaba exert a counter effect against HCD+STZ-triggered dyslipidemia.

Also, banaba or policosanol significantly elevated the HDL-C level; however, the most pronounced effect was observed following the combined supplementation with banaba+policosanol, emphasizing the role of their association in elevating HCD+STZ-reduced HDL-C levels. Policosanol has been well-documented as positively influencing HDL-C levels through various mechanisms, including cholesteryl ester transfer protein (CETP) inhibition [16,20] and upregulation of apolipoprotein A (apoA-I) [20], a key structural protein component of HDL. Additionally, previous studies suggest that corosolic acid has a modulatory effect on HDL-C [51], though the underlining mechanism remains unclear. Taken together, we hypothesize that the combination of banaba and policosanol (BNB+PCO) may synergistically enhance apoA-I expression and inhibit CETP activity more efficiently than either banaba or policosanol alone, leading to a greater improvement in HDL-C levels. Nevertheless, future comprehensive studies are necessary to confirm the notion. The present results of the blood lipid profile align with previous studies documenting the effect of policosanol [17,18] and banaba [51] in countering dyslipidemia, though no study has, so far, been conducted describing the efficacy of the banaba+policosanol combination in mitigating aggravated dyslipidemia caused by HCD+STZ. The elevated HDL-C levels in the banaba+policosanol group may be an additional reason for better tissue regeneration, as HDL-C has been closely linked to chronic wound healing [53]. Supporting this, a clinical study highlights the positive role of HDL-C in diabetic wound healing [54]. Furthermore, studies have demonstrated that the direct application of HDL-C to the wound sites significantly accelerates healing in both diabetic and non-diabetic conditions [55,56], highlighting the importance of HDL-C in tissue regeneration and wound healing.

The effect of hyperglycemia on the augmented free radicals and decreased antioxidants are the typical hallmark of diabetic conditions [57]. Perpetually, we noticed the disturbed plasma oxidative and antioxidant variables in response to the HCD+STZ treatment, such attributes being substantially restored by the consumption of banaba or policosanol. However, the most promising effect, entailing the lowest MDA levels and significantly elevated sulfhydryl content, FRA activity, and PON activity, was noticed in the banaba+policosanol-supplemented group, underscoring the synergetic positive effect of the combination against HCD+STZ-induced stress.

High cholesterol and hyperglycemia are the key contributors to liver injury and fatty liver disease [58]. The high lipid accumulation in the liver impairs its antioxidant capacity, leading to intensified oxidative stress and inflammation in the fatty liver disease [59]. During hyperglycemic and hyperlipidemic conditions, endoplasmic reticulum (ER) stress occurs in hepatocytes that modulate NFκB and c-Jun N-terminal kinase (JNK) signaling, leading to severe inflammation [59]. In the present work, we also witnessed substantial fatty liver changes, higher neutrophil counts, and a higher IL-6 production in the HCD+STZ groups, with the abovementioned phenomena prevented by the supplementation with banaba and policosanol; however, a significantly higher protection was observed when banaba and policosanol were consumed concurrently. The better effect of banaba+policosanol can be justified by their joint effect that leads to different cellular pathways converging in an effective way to prevent hepatic damage. Policosanol (Raydel^®^) also has an impact on ROS generation [16], similarly to banaba, due to its antioxidant properties [24,26] preventing oxidative stress stimulated by a variety of factors. Oxidative stress has been recognized as a major culprit in inducing apoptotic cell death [60]. In the present study, both policosanol and banaba were found to be effective in preventing oxidative stress; however, the prevention of oxidative stress was significantly higher when banaba and policosanol were used in combination (BNB+PCO), due to their distinct modes of action working synergistically. Due to the lower oxidative stress status, significantly lower apoptosis (as measured by the AO fluorescence staining) was observed in the BNB+PCO group compared to the groups supplemented with banaba or policosanol alone. The statement is supported by an earlier report which describes oxidative stress as a major provocatory event of apoptosis [60]. Also, oxidative stress has a substantial effect on the induction of senescence [61]. Herein, the BNB+PCO group exhibited reduced senescence, likely due to BNB+PCO’s enhanced ability to inhibit oxidative stress.

High liver cholesterol and hyperglycemia have adverse effects on the kidney, leading to diabetic nephropathy [62]. Policosanol is known for its kidney-protective role [63]. Similarly, few studies have documented corosolic acid’s (a main component of banaba) impact in protecting against diabetic nephropathy [64]. Herein, the significantly higher protective effect of banaba in combination with policosanol compared to their individual effect was observed against HCD+STZ-induced kidney damage. The distinct properties of policosanol (an antiglycation and hypoglycemic agent) [16,44] and banaba (a hypoglycemic, antioxidant, and anti-inflammatory agent) [23,24] work together in a synergistic manner, leading to better protective roles against kidney damage. The study outcomes are in accordance with earlier reports documenting antiglycation, antioxidant, and hypoglycemic effects as being important events to prevent diabetic nephropathy [65]. Also, kidneys from the banaba+policosanol-consuming group displayed higher efficacy in reducing HCD+STZ-induced ROS generation than the banaba or policosanol groups. The banaba+policosanol group consistently displayed the highest protective effect in mitigating the HCD+STZ-induced apoptosis and senescence. The better antiapoptotic and senescence effect can be attributed to the higher inhibitory effect of banaba+policosanol on the kidney ROS generation, which has been recognized as the inducer of apoptosis [60] and senescence [61].

## 4. Materials and Methods

### 4.1. Materials

The banaba leaf extract (batch no. UO/LSD-1825/02/23-24) was purchased from Umalaxmi Organics Pvt. Ltd. (Jodhpur, Rajasthan, India). The banaba leaves were dried, finely ground into a powder, and subjected to extraction using absolute ethanol. The ethanol extract was collected, and the solvent was removed through low-temperature heat streaming to obtain the dried powdered extract. No external additives or antioxidants were introduced during the extraction process or during the storage of the extract. The final extract was analyzed for microbial and heavy metal contamination, as well as for the quantification of its critical phytoconstituent (corosolic acid). A detailed specification and certificate of analysis of the used banaba extract is provided in the Appendix A. Policosanol (Raydel^®^, Sydney, Australia) was purified to at least >90% purity from sugarcane wax and procured from the National Center for Scientific Research (CNIC, Havana, Cuba). The used policosanol was a typical mixture of eight long-chain aliphatic alcohols (LCAA, C24-C34); a detailed specification of the policosanol product is presented in our previous report [18]. Dihydroethidium (Cat#37291), acridine orange (Cat#A9231), *N*-ε-carboxymethyllysine (Cat#14580-5g), 5-bromo-4-chloro-3-indolyl-β-D-galactopyranoside (Cat# B42525), oil red O (Cat#O0625), and 2-phenoxyethanol (Sigma P1126; St. Louis, MO, USA) were procured from Sigma Aldrich (St. Louis, MO, USA). All the other chemicals and reagents were of analytical grade and used as supplied unless otherwise stated.

### 4.2. Zebrafish Husbandry and Production of Embryos

A young zebrafish (*Danio rerio*, 16 weeks old) was cultured in a water tank equipped with a circulating water supply following the standard guidelines for Animal Use and Care adopted by the Raydel Research Institute (approval code RRI-23-2007, approval date 27 July 2023). The zebrafish were fed normal terabit (ND) and maintained under alternating light (14 h) and dark (10 h) photo periods.

To produce embryos, male and female zebrafish were placed in the breeding tank and separated from each other using the perforated physical divider. After 16 h, the divider was removed, allowing the male and female zebrafish to breed uninterruptedly. Following 30 min of mating, the embryos were collected, washed with water, and used for further study.

### 4.3. Microinjection of Zebrafish Embryos

Zebrafish embryos [1.5 h post fertilization (hpf)] were segregated into four different groups (n = 100/group). Group I embryos were microinjected (10 nL) with phosphate-buffered saline (PBS), while the embryos in group II were injected with 500 ng of CML/10 nL of PBS. Group IV and V embryos received 5 ng and 10 ng of banaba suspended in 10 nL of PBS containing 500 ng of CML. Microinjection was performed under a stereomicroscope (Motic SMZ 168; Hong Kong, China) using microcapillary needles and a pneumatic picopump (PV8830; World Precision Instruments, Sarasota, FL, USA) equipped with a magnetic manipulator (MM33; Kanetc, Bensenville, IL, USA). Embryos in the different groups were monitored under a stereomicroscope (Motic SMZ 168; Hong Kong, China) up to 72 h post injection to examine their survivability, hatching, and developmental deformities. The selection of the 5 ng and 10 ng doses of banaba was based on preliminary experiments where banaba doses ranging from 0 to 25 ng were injected into zebrafish embryos in the presence of CML (500 ng) following a survivability analysis 48 h post injection. The results revealed a protective effect of banaba that significantly improved with the increasing concentration of banaba up to 10 ng. At higher doses (15, 20, and 25 ng), the effect was similar and was comparable to that observed under the 10 ng dose. Consequently, the 10 ng dose of banaba was selected as the optimal high dose for the microinjection studies, while the 5 ng dose was selected to assess the dose-dependent effect.

### 4.4. Acute Toxicity in Adult Zebrafish

Acute toxicity in adult zebrafish (16 weeks old) was induced through an intraperitoneal injection of 250 μg of CML (equivalent to 3 mM)/10 μL of 50% ethanol in PBS. Zebrafish (n = 30) were randomly allocated to three different groups. Zebrafish (n = 10) were injected with 10 μL of PBS (PBS alone group). Similarly, zebrafish (n = 10/group) were injected with 250 μg of CML suspended in 10 μL of 50% ethanol in PBS (3 mM CML + 50% EtOH group) and 250 μg of CML suspended in 10 μL of 50% ethanol in PBS containing 8 μg of the banaba extract (3 mM CML+50% EtOH+banaba group). Among all the groups, the intraperitoneal injection was given by using a 28-gauge needle after anaesthetizing the zebrafish by drenching them in 2-phenoxyethanol (0.1% *v*/*v*). The zebrafish were constantly monitored until 90 min post injection to assess survivability and swimming activity following the OECD 2019 guidelines [66].

### 4.5. Blood Analysis for Malondialdehyde (MDA) Levels and Antioxidant Variables

The zebrafish from all the groups were sacrificed 90 min post injection using the hypothermic shock method [67], and blood was immediately collected. The oxidative status of the plasm was assessed by quantifying the malondialdehyde (MDA) level using a thiobarbituric acid reactive substance (TBARS) assay [68]. In brief, 1 mg/mL equivalent plasma was suspended in 20 μL of PBS and subsequently blended with 50 μL of trichloroacetic acid (20%) and 100 μL of thiobarbituric acid (0.67%) and incubated at 95 °C for 10 min. Finally, an absorbance of 560 nm was recorded, and the results were quantified by using the MDA standard (0.5–50 μM).

The plasma antioxidant variables were determined by assessing the ferric ion reduction ability (FRA), the sulfhydryl content, and the paraoxonase (PON) activity. The plasma FRA was determined by the method described earlier, with slight amendments [68]. Briefly, 1 mg/mL protein equivalent plasma was dissolved in 20 μL of PBS and mixed with 180 μL of FRA reagents [68]. Following 1 h of incubation at room temperature, an absorbance of 593 nm was recorded.

The sulfhydryl content was assessed using the 5,5-dithiol-bis-(2-nitrobenzoic acid) (DTNB) method [68]. Briefly, 50 μL of plasma (1.0 mg/mL protein) was mixed with 50 μL of DTNB (4 mg/mL). Following 2 h of incubation at room temperature, an absorbance of 412 nm was recorded to measure the chromogenic product 5-thiol-2-nitrobenzoic acid (TNB), with the results expressed as nmol/mg of protein employing the 13.6 × 10^3^ M^−1^ cm^−1^ molar absorbance coefficient (ε) of TNB. 

Paraoxonase (PON) activity was assessed by mixing 40 μL of plasma (1.0 mg/mL equivalent protein) with 160 μL of paraoxon ethyl (0.15 g/mL) [68]. Following a 2 h incubation at 25 °C, an absorbance of 415 nm was recorded to quantify the formed product *p*-nitrophenol, and the PON activity was expressed as μU/L/min using the 1.7 × 10^4^ M^−1^ cm^−1^ molar absorbance coefficient (ε) of *p*-nitrophenol. 

The liver function biomarkers in the plasma, i.e., aspartate aminotransferase (AST) and alanine aminotransferase (ALT), were assessed using the commercial detection kit, following the manufacturers’ guidelines; a detailed methodology is provided in the Appendix A.

### 4.6. Formulation of Different Diets

The normal diet (ND), tetrabit, a regular diet of zebrafish, was mixed with cholesterol (final 4%, *w*/*w*) to make a high-cholesterol (HCD) diet. The HCD was infused with banaba leaf extract (final 0.1% *w*/*w*), policosanol (final 0.1% *w*/*w*), and 0.1% each of banaba and policosanol to make three different dietary formulations named HCD+BNB, HCD+PCO, and HCD+BNB+PCO, respectively. The 0.1% dose was selected based on our previous study [69], where 0.1% policosanol displayed the optimal effect against high-cholesterol diet-induced dyslipidemia and associated adverse events. Furthermore, for the comparative evaluation, a similar amount of banaba (i.e., 0.1%) was used.

### 4.7. Induction of Hyperglycemia in Zebrafish and Tail Fin Regeneration

Adult zebrafish (18 weeks old, n = 60) were fed exclusively with HCD for 8 weeks (56 days) to induce hyperlipidemia following tail fin amputation using the earlier described method [70]. In brief, the zebrafish were anaesthetized by submerging them in a 0.1% 2 phenoxyethanol solution, followed by tail amputation using a sterilized surgical blade near the dermal rays of the tail fin. The tail-fin-amputated hyperlipidemic zebrafish (n = 60) were randomly divided into five groups (n = 12/group) (Figure 10). The zebrafish in group I were exclusively fed with HCD. At the same time, the zebrafish in groups II to V received the four injections of streptozotocin (STZ, 10 μL, final 0.9%) at one-day intervals and were fed with HCD alone (HCD+STZ group), HCD+banaba (HCD+STZ+BNB group), HCD+policosanol (HCD+STZ+PCO group), and HCD+banaba+policosanol (HCD+STZ+BNB+PCO group). Tail fin regeneration across all groups was measured on days 0, 4, 6, 10, and 14 under a stereomicroscope connected with a digital camera, and the tail fin regeneration area was quantified using the Image J software (https://imagej.net/ij, 1.53 version, accessed on 6 June 2023).

### 4.8. Blood Analysis and Collection of Organs

After 14 days of intervention with different diets, the zebrafish (group I–V) were sacrificed by hypothermic shock and blood was immediately collected. The liver and kidney from the different groups were surgically exiced and preserved in 10% formalin for the histological analysis.

Blood from the different groups was analyzed for the glucose level using a blood detection kit (ASAN Pharma. Co., Ltd., Hwaseong, Republic of Korea) following the manufacturer’s guidelines. The total blood cholesterol (TC), triglycerides (TGs), and high-density lipoproteins (HDL-C) were quantified by employing the commercial lipoprotein quantification kits following the method suggested by the manufacturers. The liver function biomarkers aspartate aminotransferase (AST) and alanine aminotransferase (ALT) were assessed using the commercial detection kit following the guidelines provided by the manufacturers. A detailed methodology is provided in the Appendix A.

Blood MDA, sulfhydryl content, and FRA and PON activity were assessed using the methodology described in Section 4.5.

### 4.9. Histological Analysis and Cellular Senescence

The tissues (liver and kidney) were fixed in the FSC 22 clear frozen solution (Leica) and processed for cryo-sectioning using the cryo-microtome (Leica CM1510 S, Leica biosystem, Nussloch, Germany) to obtain a tissue section (7 μm thick). The tissue section was processed for hematoxylin and eosin (H&E) staining to assess the histological changes in the liver and kidney.

Fatty liver changes in the hepatic section were examined by oil red O (ORO) staining [67]. In brief, the hepatic tissue section was stained with an ORO solution (0.25 mL, 0.3%). After incubation under mild heating conditions (5 min, 60 °C), the stained section was rinsed thoroughly with water and examined under a microscope.

Cellular senescence was detected using the senescence-associated β galactosidase (SA-β-gal) assay [71]. In brief, the tissue section was stained with a 5-bromo-4-choloro-3-indolyl- β-D-galactopyranoside (X-gal, 0.1%) solution [71]. The tissue section was visualized under a microscope for the detection of blue-stained senescence-positive cells after 16 h of staining.

### 4.10. Immunohistochemistry (IHC) and Fluorescence Imaging

The interleukin (IL)-6 was detected in the hepatic section using IHC [72]. The tissue section was flooded with the 200× time diluted anti-IL-6 antibody (ab9324, Abcam, London, UK). Following 16 h of incubation under cool conditions, the section was developed using an EnVisin+system HRP polymer kit (Code K4001, Dako, Glostrup, Denmark).

For fluorescence imaging to detect ROS generation and apoptosis, a tissue section (7 μm) was stained with dihydroethidium (DHE, 30 μM) and acridine orange (AO, 5 μg/mL) [67]. Following 30 min of incubation in the dark, the tissue section was visualized using fluorescence microscopy at the excitation/emission wavelength of 585/615 nm for DHE and 505/535 nm for AO.

### 4.11. Statistical Analysis

One-way and two-way analyses of variance (ANOVA), followed by Tukey’s post-hoc analyses, were conducted using the SPSS software (version 29.0; Chicago, IL, USA) to establish the statistical difference between the multivariate groups. The two-tailed Student’s *t*-test was utilized to establish pairwise statistical differences between the groups. All the data depicted in the graphs were obtained from the triplicate experiments and are represented as the mean ± SEM.

## 5. Conclusions

Both banaba and policosanol displayed substantial effects in alleviating hyperglycemia and dyslipidemia in diabetic hyperlipidemic zebrafish. However, these effects accelerated greatly in a synergistic manner through the combination of banaba and policosanol. Similarly, the combination of banaba+policosanol prompted tissue regeneration and had a substantial inhibitory effect on the oxidative variables via the augmentation of the antioxidant parameters. Due to the substantial hypoglycemic effect and antioxidant property combination of banaba+policosanol, the latter effectively protected against fatty liver changes and hepatic IL-6 generation in the diabetic hyperlipidemic zebrafish. Likewise, the banaba+policosanol supplementation safeguarded the kidney from oxidative stress, apoptosis, and cellular senescence actuated by the diabetic hyperlipidemic conditions. The findings confirm the superior efficacy of the banaba+policosanol combination compared to their individual supplementation in alleviating the metabolic stress induced by diabetic hyperlipidemic conditions. As a future prospect, studies will be conducted to compare the effects of the banaba+policosanol combination with standard antidiabetic drugs. Additionally, genomics and proteomics studies will be explored to decode the protective molecular mechanism exerted by banaba+policosanol. Given the promising preclinical findings in zebrafish, a human clinical trial will be planned to further assess the potential efficacy of banaba+policosanol in managing diabetic hyperlipidemic conditions.

## Figures and Tables

**Figure 1 pharmaceuticals-18-00362-f001:**
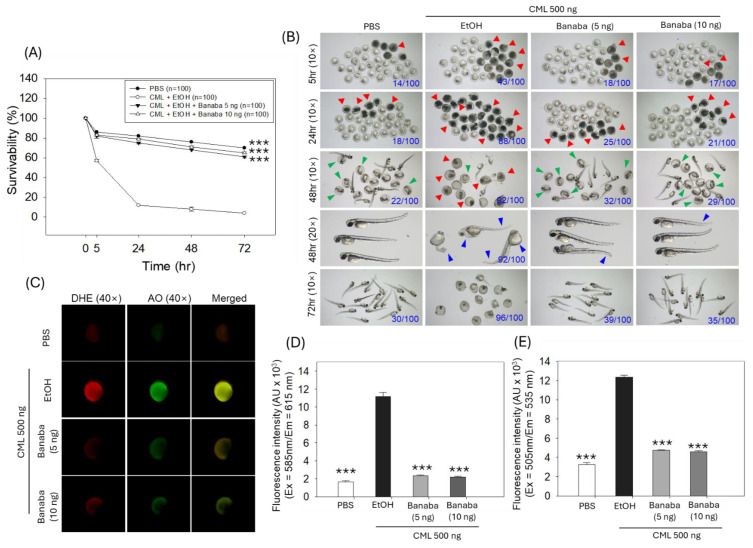
Effect of banaba leaf extract against carboxymethyllysine (CML)-induced toxicity in zebrafish embryos (n = 100). (**A**) The embryo survivability kinetics at two different concentrations (5 ng or 10 ng banaba/embryo). The symbol *** (*p* < 0.001) indicates the statistical significance of the difference between groups vs. EtOH+CML-injected group as determined by a two-way ANOVA, followed by Tukey’s post-hoc analysis. (**B**) Pictorial representation of zebrafish embryos among the different groups 5 h to 72 h post injection. Blue numerical values inside the images represent the number of dead embryos/total embryos (n = 100). The red arrowhead highlights the dead embryos, the green arrowhead represents the unhatched embryos 48 h post injection, and the blue arrowhead underscores the developmental deformities. (**C**) Dihydroethidium (DHE) and acridine orange (AO) fluorescent imaging of the embryos. (**D**,**E**) Quantification of DHE and AO fluorescence intensity using the Image J software. Abbreviations: PBS (phosphate buffered saline), EtOH (ethanol), CML (carboxymethyllysine). The symbol *** signifies the statistical difference at *p* < 0.001 vs. EtOH+CML-injected group.

**Figure 2 pharmaceuticals-18-00362-f002:**
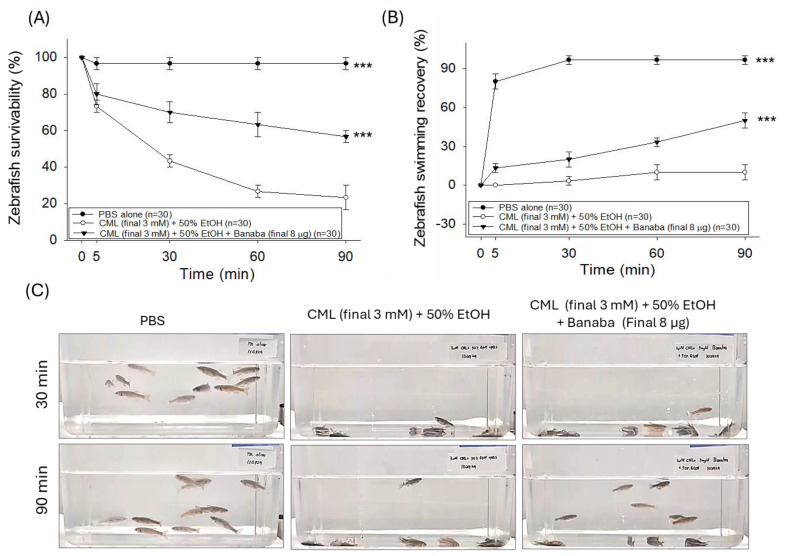
Effect of banaba leaf extract against the carboxymethyllysine (CML)-induced toxicity in adult zebrafish (n = 30). (**A**) Kinetics of zebrafish survivability 90 min post treatment. (**B**) Percentage of swimming recovery in adult zebrafish 90 min post treatment. (**C**) Pictorial view of the zebrafish swimming 60 min and 90 min post treatment. The symbol *** (*p* < 0.001) indicates the statistical significance of the difference between groups vs. the group injected with CML (final 3 mM)+50% EtOH as determined by two-way ANOVA, followed by Tukey’s post-hoc analysis.

**Figure 3 pharmaceuticals-18-00362-f003:**
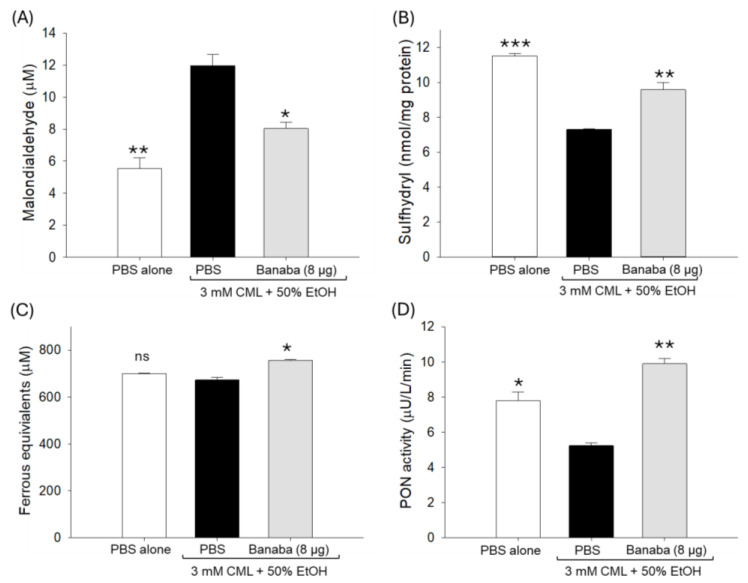
The plasma antioxidant status of adult zebrafish injected with carboxymethyllysine (CML) with or without the banaba leaf extract. (**A**) Quantification of the lipid peroxidation assessed by thiobarbituric acid reactive substance (TBARS) assay using malondialdehyde (MDA) as the standard. (**B**) Ferric ion reduction assay (FRA). (**C**) Sulfhydryl content quantification. (**D**) Paraoxonase (PON) activity assay. Each analysis was performed using triplicate experiments with duplicate samples, resulting in a total of six replicates (n = 6). * (*p* < 0.01), ** (*p* < 0.01), and *** (*p* < 0.001) displayed the statistical difference between the groups with respect to the group injected with 3 mM CML+50% EtOH; ns represents a non-significant (*p* > 0.05) difference.

**Figure 4 pharmaceuticals-18-00362-f004:**
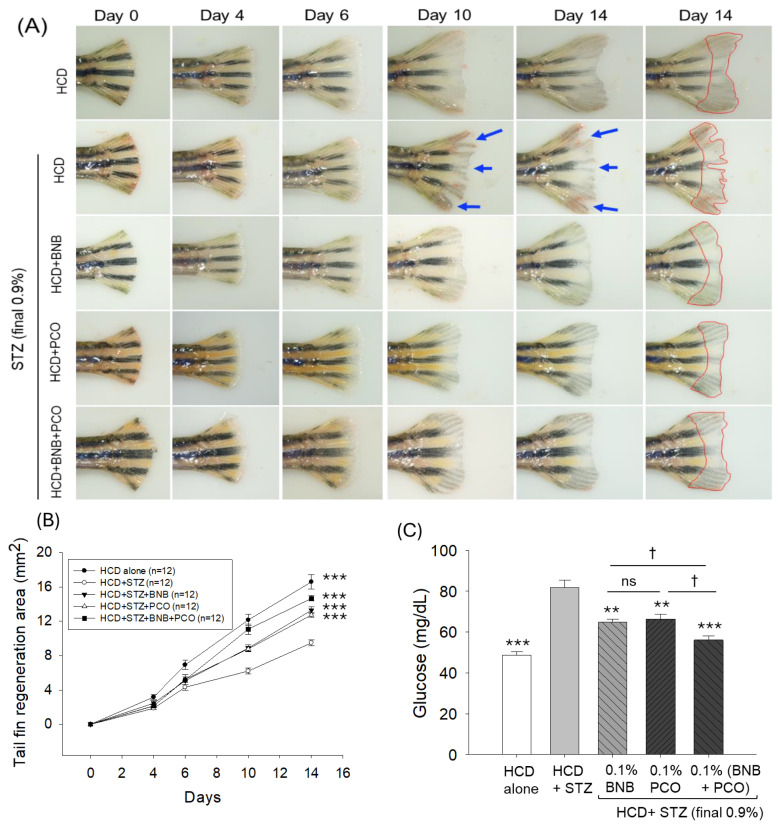
Effect of banaba leaf extract and policosanol (alone or in combination) supplementation on the tail fin regeneration and blood glucose level of the hyperlipidemic zebrafish injected with streptozotocin (n = 12). (**A**) Pictorial view of amputated tail fin regeneration (tissue regeneration) after 14 days. The blue arrow indicates the tail fissure, while the red line depicts the regenerated tail fin area after 14 days. (**B**) A time-dependent (0–14 days) tail fin regenerated area was observed among the different treatment groups. The symbol *** (*p* < 0.001) indicates the statistical significance of the difference between groups vs the group injected with EtOH+CML, as determined by the two-way ANOVA, followed by Tukey’s post-hoc analysis. (**C**) The blood glucose level was examined after 14 days. Abbreviations: HCD (high cholesterol diet), STZ (streptozotocin), BNB (banaba), and PCO (policosanol). The symbols **, and *** underline the statistical significance at *p* < 0.01, and 0.001, respectively, vs. the HCD+STZ group. † indicate statistical significance at *p* < 0.05 in relation to the BNB+PCO group; ns represents a non-significant (*p* > 0.05) difference.

**Figure 5 pharmaceuticals-18-00362-f005:**
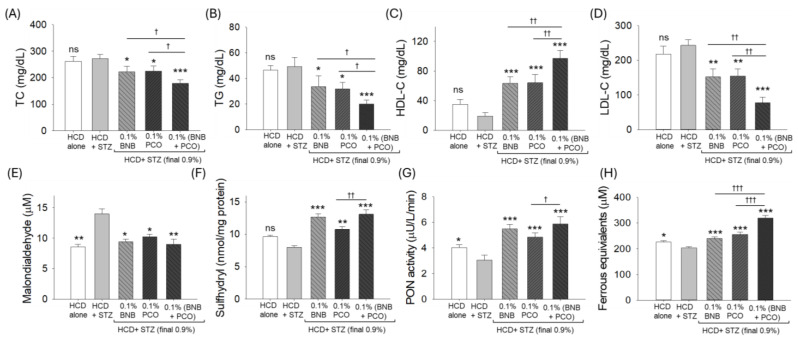
Effect of banaba leaf extract and policosanol (alone or in combination) on the lipoprotein profile and oxidative and antioxidant variables of plasma from the hyperlipidemic zebrafish treated with streptozotocin (n = 12). (**A**) Total cholesterol (TC), (**B**) triglycerides (TGs), (**C**) high-density lipoprotein cholesterol (HDL-C), (**D**) low-density lipoprotein cholesterol (LDL-C), (**E**) malondialdehyde (MDA), (**F**) sulfhydryl content, (**G**) paraoxonase (PON), and (**H**) ferric ion reduction (FRA) ability. Each analysis was performed using triplicate experiments with duplicate samples, resulting in a total of six replicates (n = 6). Abbreviations: HCD (high cholesterol diet), STZ (streptozotocin), BNB (banaba), and PCO (policosanol). The symbols *, **, and *** underline the statistical significance at *p* < 0.05, *p* < 0.01, and *p* < 0.001, respectively, vs. the HCD+STZ group. †, ††, and ††† indicate statistical significance at *p* < 0.05, *p* < 0.01, and 0.001 vs. the BNB+PCO group; ns represents a non-significant (*p* > 0.05) difference vs. the HCD+STZ group.

**Figure 6 pharmaceuticals-18-00362-f006:**
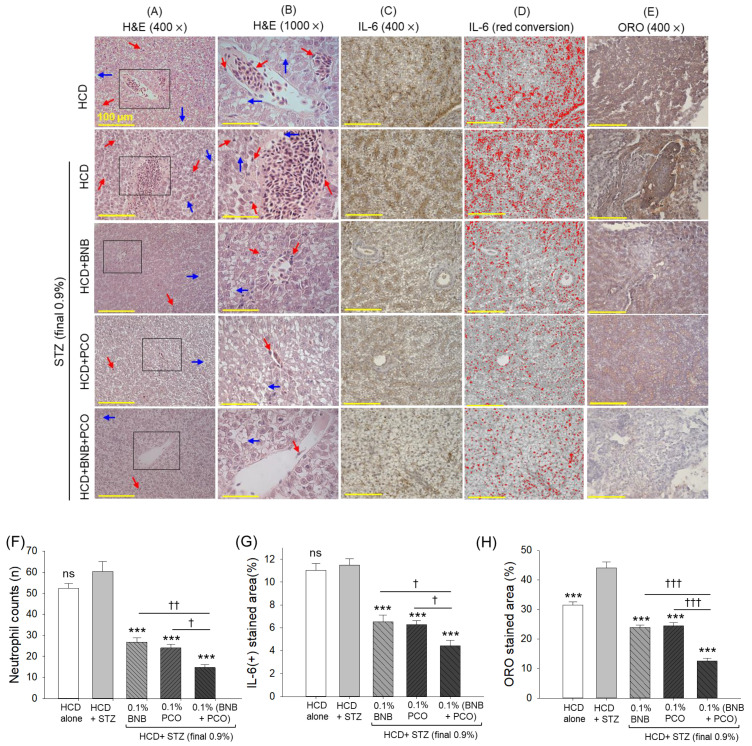
A comparative effect of banaba leaf extract and policosanol (alone or in combination) on the hepatic histology of the streptozotocin-treated hyperlipidemic zebrafish (n = 3). (**A**) Hematoxylin and eosin (H&E) staining (400× magnified): blue and red arrows highlight the infiltration of neutrophils and lipid droplets, respectively [100 μm, scale bar]. The H&E area in the black box is 1000× magnified and images (**B**) represent this magnified view. (**C**) Oil red O (ORO) staining. (**D**) Immunohistochemical (IHC) staining for the detection of interleukin (IL)-6. (**E**) Red conversion of the IL-6-stained area (brown color) using the Image J software with a brown color threshold value of 20–120; the red conversion was performed to improve the intensity of the IL-6-stained area. (**F**) Percentage neutrophil counts. A semiquantitative evaluation of neutrophils (stained with a dark purple color) was conducted through a microscopic analysis of a predefined region (1.23 mm^2^) across three independent sections, with five distinct areas assessed per section. (**G**,**H**) illustrate the quantification of ORO and IL-6-stained areas, respectively. Abbreviations: HCD (high cholesterol diet), STZ (streptozotocin), BNB (banaba), and PCO (policosanol). The symbols *** underline the statistical significance at *p* < 0.001vs the HCD+STZ group. †, †† and ††† indicate statistical significance at *p* < 0.05, *p* < 0.01 and *p* < 0.001 vs. the BNB+PCO group; ns represents a non-significant (*p* > 0.05) difference.

**Figure 7 pharmaceuticals-18-00362-f007:**
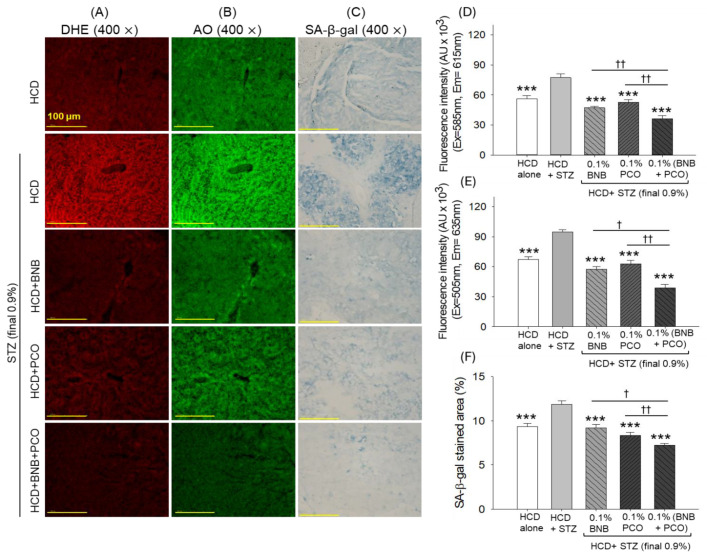
Effect of banaba leaf extract and policosanol (alone or in combination) on the reactive oxygen species (ROS), apoptosis, and senescence in the liver of the streptozotocin-treated hyperlipidemic zebrafish (n = 3). (**A**,**B**) dihydroethidium (DHE) and acridine orange (AO) fluorescent staining to determine ROS and apoptosis, respectively. (**C**) Senescence-associated β-galactosidase (SA-β-gal) staining. (**D**,**E**) Quantification of DHE and AO fluorescence intensity. (**F**) Percentage quantification of the SA-β-gal-positive area. Abbreviations: HCD (high cholesterol diet), STZ (streptozotocin), BNB (banaba), and PCO (policosanol). The symbol *** underlines the statistical significance at *p* < 0.001 vs. the HCD+STZ group. † and †† indicate statistical significance at *p* < 0.05 and *p* < 0.01 vs. the BNB+PCO group.

**Figure 8 pharmaceuticals-18-00362-f008:**
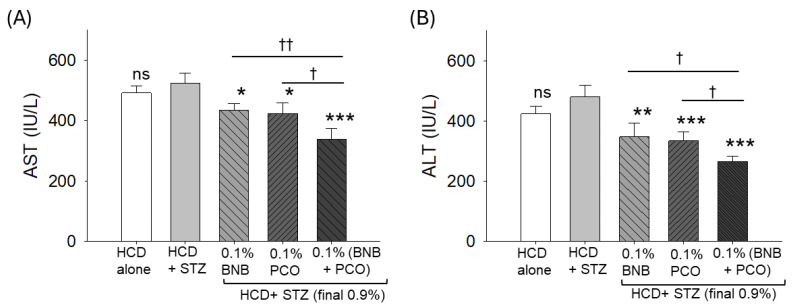
Effect of banaba leaf extract and policosanol (alone or in combination) on the blood hepatic function biomarkers in streptozotocin-treated hyperlipidemia zebrafish (n = 12). (**A**) Aspartate amino transferase (AST) and (**B**) alanine amino transferase (ALT). Each analysis was performed using triplicate experiments with duplicate samples, resulting in a total of six replicates (n = 6). Abbreviations: HCD (high cholesterol diet), STZ (streptozotocin), BNB (banaba), and PCO (policosanol). The symbols *, ** and *** underline the statistical significance at *p* < 0.05, *p* < 0.01 and *p* < 0.001, respectively, vs. the HCD+STZ group. † and †† indicate statistical significance at *p* < 0.05 and *p* < 0.01 vs. the BNB+PCO group; ns represents a non-significant (*p* > 0.05) difference.

**Figure 9 pharmaceuticals-18-00362-f009:**
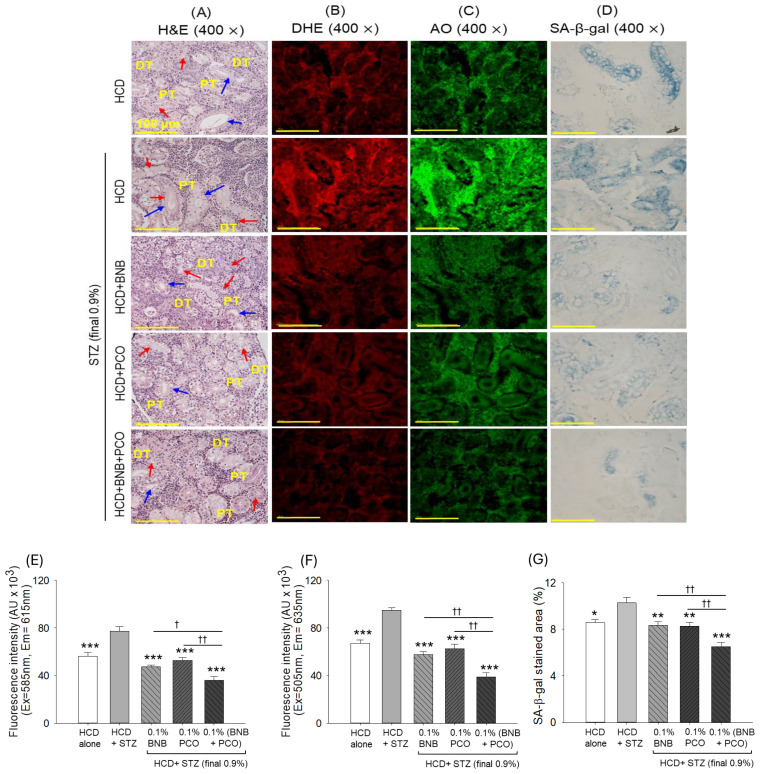
Effect of banaba leaf extract and policosanol (alone or in combination) on the kidney histology of the streptozotocin-treated hyperlipidemia zebrafish (n = 3). (**A**) Hematoxylin and eosin (H&E) staining. PT and DT represent the proximal and distal tubules; the blue and red arrows describe the enlarged tubular lumen and luminal debris. (**B**) Dihydroethidium (DHE) and (**C**) acridine orange (AO) fluorescent staining were used to determine ROS and apoptosis, respectively. (**D**) Senescence-associated β-galactosidase (SA-β-gal) staining. (**E**,**F**) illustrate the quantification of DHE and AO fluorescence intensity, respectively. (**G**) Percentage quantification of the SA-β-gal-positive area. Abbreviations: HCD (high cholesterol diet), STZ (streptozotocin), BNB (banaba), and PCO (policosanol). The symbols *, **, and *** underline the statistical significance at *p* < 0.05, *p* < 0.01, and *p* < 0.001, respectively, vs. the HCD+STZ group. † and †† indicate statistical significance at *p* < 0.05 and *p* < 0.01 vs. the BNB+PCO group.

**Figure 10 pharmaceuticals-18-00362-f010:**
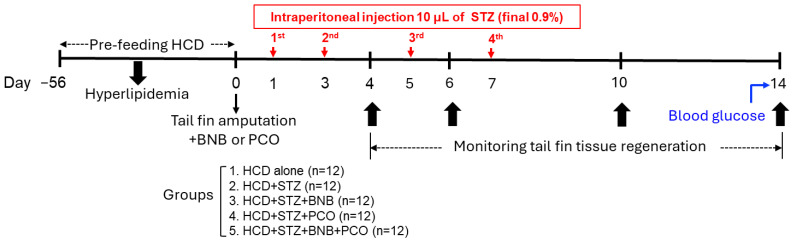
Study layout of the development of hyperlipidemia in adult zebrafish with and without hyperglycemia induction by streptozotocin and tail fin regeneration. Abbreviations: HCD (high cholesterol diet), STZ (streptozotocin), BNB (banaba leaf extract), and PCO (policosanol). HCD was prepared by mixing cholesterol and tetrabits, a brand name of the zebrafish diet which was purchased from Tetrabit Gmbh, Melle, Germany, and contains 47.5% crude protein, 6.5% crude fat, 2.0% crude fiber, and 10.5% crude ash, as well as vitamin A (29,770 IU/kg), vitamin D3 (1860 IU/kg), vitamin E (200 mg/kg), and vitamin C (137 mg/kg).

## Data Availability

The data used to support the findings of this study are available from the corresponding author upon reasonable request.

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
