# Peer review of "Synergistic Efficacy of Policosanol (Raydel®) and Banaba Leaf Extract to Treat Hyperglycemia and Dyslipidemia in Streptozotocin-Induced Diabetic and Hyperlipidemic Zebrafish (Danio rerio): Protection of Liver and Kidney with Enhanced Tissue Regeneration"

_pharmaceuticals, 2025, doi:10.3390/ph18030362_

Round 1
Reviewer 1 Report
Comments and Suggestions for Authors
In this study, Cho et al studied the synergistic efficacy of policosanol and banaba extract to treat hyperglycemia and dyslipidemia. The study is of interest, however, there are concerns to be addressed.
Major concerns:
1. The statistical method is questionable. Is the test two-tailed or one-tailed?
2. Only one-way ANOVA and t-test were mentioned. However, they are not the correct test for Figure 1A, Figure 2A and 2B, and Figure 4B.
3. Figure 6 F: HE staining for neutrophil quantification is questionable. Suggest to repeat with IHC staining.
Minor concerns:
1. Line 127: Explain the reasons for choosing the banaba doses of 5 ng and 10 ng
2. Line 149: Figure 1, need to state the sample size
3. Line 181: Figure 2, need to state the sample size
4. Line 204: Figure 3, need to state the sample size
5. The same for other figures to state the sample size
6. Line 282: Change “ns represents non-significant (p>0.05) difference” to “ns represents non-significant (p>0.05) difference vs HCD+STZ group”
Author Response
Thank you for your insightful comments. Following the reviewer’s suggestion, we made point-to-point response and reflected on revision.
Please find attached doc as our response.

Reviewer 2 Report
Comments and Suggestions for Authors
The research article pharmaceuticals-3474098 is of interest and is particularly relevant in light of the rapid increase in the incidence of metabolic diseases, such as hypercholesterolemia and type II diabetes, as well as the need to explore natural alternatives to synthetic drugs, which patients often take for extended periods and can have significant side effects. The article is comprehensive, and the results could potentially be published in two separate papers. The research appears to be well-conducted and meticulously detailed. The authors demonstrate a solid understanding of the subject. I have no substantive objections to the work, but the manuscript would benefit from additional data. I have several comments, which I detail below.
- The authors did not write what family the plant banaba belongs to, whether it is an annual or a perennial. Is it cultivated or does it grow wild? Two succinct sentences here would have been helpful.
- Based on the articles [23] and [25], and perhaps others, I would try to explain in more detail than the phrase: “(...) insulin secretion, insulin sensitization/resistance and inhibition of enzymes such as α-amylase and α-glucosidase [23,25],” what are the molecular mechanisms of action of the compounds contained in banaba leaf extract for its antidiabetic effects.
- The extract was described as follows: “Banaba leaf extract was purchased from Umalaxmi Organics Pvt. Ltd (Jodhpur, Rajasthan, India)”. In my opinion, this information is completely inadequate, especially since it does not refer to any work done on this particular extract. There is no citation here. I conclude that this is an ethanol extract, am I correct? Is anything known about the extraction itself? Were the leaves used for it fresh, dried or freeze-dried? I am a bit puzzled why the authors did not make the extracts themselves. Are they now sure that the extract did not contain any additives, such as stabilizers or antioxidants? Are they sure that it was stored in a way that ensured its maximum biological activity before it got into their hands after purchase?
- The drug/dietary supplement “Cuban policosanol (Raydel®)” appeared in the title of the paper. However, throughout the manuscript, the term “Cuban” appears nowhere else. In the description of the materials, the authors listed “Policosanol (Raydel®, Sydney, Australia).” Policosanol is a mixture produced and sold in many countries, so I do not understand why the word “Cuban” appeared in the title. A lesser surprise would be the term “Australian,” although it is generally not customary to describe a manufacturer's country in this way in scientific articles, especially in the title and especially of substances that are not unique in any way.
- It would be beneficial to include more information in the article about the future prospects of the study, i.e. what further step can be taken to bridge the gap between research in a model such as zebrafish and clinical trials.
Editorial:
- When giving the name of zebrafish for the first time, it should be supplemented with the Latin name (Danio rerio).
- Line 153: The name 'banaba' should be written starting with a lowercase letter.
- Line 198: The abbreviations FRA and PON appear here for the first time, so they need to be explained (these explanations appear later in the paper, where it can be considered whether they are necessary).
- Line 256: There is an error in notation 19.2.2.
- Line 660: In the phrase “incubation at 25oC”, a proper degree symbol should be used.
- In the list of literature, the doi numbers of individual publications should be completed.
- In the Supplementary File, there is a sentence: “The plasma total cholesterol (TC) and triglyceride (TG) were determined using commercial assay kits (cholesterol, T-CHO, and TGs, Cleantech TS-S; Wako Pure Chemical, Osaka, Japan)”. I would write triglycerides (plural) and use the same abbreviation, i.e., either TG or TGs throughout the work.
- In the Supplementary File, in the sentence: “Similarly, 5 μL serum was mixed with a 200 μL TG-specific reaction mixture (…)” the prepositions "of" are missing.
Feb. 7, 2025
Author Response

(The authors gave the same response as above.)

Reviewer 3 Report
Comments and Suggestions for Authors
Dear Authors.
I have carefully studied your scientific article: a very detailed and extremely interesting scientific work has been submitted for consideration, which goes far beyond the stated objectives.
The combination of banaba+policosanol prompted tissue regeneration and had a substantial inhibitory effect on the oxidative variables via the augmentation of the antioxidant parameters. Due to the substantial hypoglycemic effect and antioxidant property combination of banaba+policosanol effectively protected the fatty liver changes and hepatic IL-6 generation in the diabetic hyperlipidemic zebrafish. Likewise, the banaba+policosanol supplementation safeguarded the kidney from oxidative stress, apoptosis and cellular senescence actuated by the diabetic hyperlipidemic conditions.
Without a doubt, the article is worthy of being published in such a respected Journal, but I would like to clarify some inaccuracies and possible errors during a detailed study of the text:
- Authors write: The most impressive impact of BNB+PCO was noticed in the elevation of High-density lipoprotein cholesterol that was 1.5 fold higher than the HDL-C level in response to BNB and PCO. Additional explanation of the effect obtained in the experiments is required. Why such a result? It is possible to provide additional scientific literature in the text for clarification.
- Authors write: A similar trend was observed for apoptosis where the HCD+STZ induced higher AO fluorescent intensity, which was effectively reduced by ..............following the supplementation of policosanol, Banaba and the combination of banaba+policosanol , respectively. Similar question: a statement of the obtained results is presented. Additional explanation of the effect obtained in the experiments is required. Why such a result? More detailed explanations of the obtained effect are possible in the Discussion section with additional scientific literature.
- Authors postulated: The combined study obtained from embryos and adult zebrafish underscores the protective effect of Banaba in mitigating CML-posed oxidative steps, ascertained by its potent antioxidant capabilities. It is unclear whether this phrase refers to the studies cited in the article or to some other studies? Then a link is needed or the phrase needs to be reformatted.
Author Response

(The authors gave the same response as above.)

Reviewer 4 Report
Comments and Suggestions for Authors
Authors studied protective effects of banaba leaf extract (BNB) and policosanol (PCO) on CML-induced toxicity and hyperlipidemic diabetes in zebrafish models. The study revealed that NB and PCO, both individually and in combination, effectively mitigate metabolic and inflammatory damage associated with dyslipidemia and hyperglycemia.
Authors utilized both embryo and adult zebrafish models, which provides a comprehensive perspective on the biological effects of BNB and PCO at different life stages.
The BNB+PCO combination shows a significantly greater therapeutic benefit than either compound alone. The improvements in blood glucose control, lipid profile, oxidative stress, and inflammatory markers highlight potential clinical applications for managing diabetes and metabolic disorders.
Yet, the study must be revised. My recommendations are as follows:
- Authors established that BNB and PCO exert beneficial effects, but the manuscript lacks detailed mechanistic insight into how these compounds interact with molecular pathways. Future studies could incorporate gene expression analysis or proteomics to understand how BNB and PCO regulate metabolic and inflammatory pathways at a molecular level.
- BNB and PCO showed therapeutic effects in the present work, however, the specific active compounds responsible for these benefits were not thoroughly discussed. Identifying bioactive molecules (e.g., corosolic acid in BNB, long-chain alcohols in PCO) would enhance the study’s pharmacological relevance.
- Authors did not compare BNB and PCO with existing diabetes or lipid-lowering treatments (e.g., metformin, statins). Including a drug control group would help determine whether BNB and PCO provide comparable or superior benefits to current pharmacological treatments. Discuss and acknowledge please.
- Authors used a fixed dose (0.1% w/w) for BNB and PCO, but dose-response analyses would help determine the optimal concentration for maximum therapeutic benefit.
- While zebrafish are a widely used metabolic model, their glucose metabolism differs from mammals, and findings should be cautiously extrapolated to humans. Discuss please.
Author Response

(The authors gave the same response as above.)

Round 2
Reviewer 1 Report
Comments and Suggestions for Authors
Thank for addressing my concerns.
Reviewer 4 Report
Comments and Suggestions for Authors
Authors look revised the paper properly. No more issues in the text. Hence, I recommend publication of the manuscript in the journal.